# Expression of *Pleurotus ostreatus* Laccase Gene in *Pichia pastoris* and Its Degradation of Corn Stover Lignin

**DOI:** 10.3390/microorganisms8040601

**Published:** 2020-04-21

**Authors:** Qian Song, Xun Deng, Rui-Qing Song

**Affiliations:** 1Northeast Forestry University, Forestry College, Forest Protection Discipline, Harbin 150040, China; 961725342@nefu.edu.cn; 2Heilongjiang Academy of Forestry, Institute of Forest Protection, Harbin 150040, China; dxhappy@126.com

**Keywords:** *Pleurotus ostreatus*, laccase, *Pichia pastoris*, lignin, degradation

## Abstract

*Pleurotus ostreatus* is a species of white-rot fungi that effectively degrades lignin. In this study, we aimed to efficiently express the *lac-2* gene of *Pleurotus ostreatus* in the *Pichia pastoris* X33 yeast strain. The enzymatic properties of recombinant yeast were determined, and its ability to degrade corn stover lignin was determined. The results showed the optimum pH values of recombinant laccase for 2,2’-Azinobis-3-ethylbenzothiazoline-6-sulfonic acid, 2,6-dimethoxyphenol, and 2-methoxyphenol were 3.0, 3.0, and 3.5, respectively. The optimum reaction temperature was 50 °C, and it had good thermal stability and acid and alkali resistance. The degradation rate of lignin in corn stover by recombinant laccase was 18.36%, and the native *Pleurotus ostreatus* degradation rate was 14.05%, the difference between them is significant (*p* < 0.05). This experiment lays a foundation for the study of the degradation mechanism of lignin by laccase.

## 1. Introduction

Laccases (EC 1.10.3.2) are polyphenol copper blue proteases containing four copper ions (except *Phlebia radiata* laccase, which has two copper ions) and exist in plants, fungi, insects, and a small number of bacteria [1,2,3,4,5]. Since Yoshida [6] discovered laccase 131 in lacquer sap in 1883, laccase has become one of the hot research topics in the chemical industry, biological sciences, and environmental science [7,8,9]. During the catalytic process, the interaction between the four copper ions in the laccase catalytic core causes electron transfer and self-level change, reducing oxygen to water and oxidizing the substrate molecule to generate free radicals. Finally, the active intermediates are transformed into dimers, oligomers, and polymers [10,11,12].

In recent years, it has been found that the application value of laccase is very high, as it can be used for paper bleaching, medical and health purposes, as bio-fertilizer, and to degrade stubborn pollutants. Among these uses, the degradation of environmental pollutants by laccase has been widely studied [13,14]. Laccase can degrade the complex structure of lignin that makes up the cell wall of stover, which makes it an excellent candidate to biodegrade renewable biological resources for reuse. However, the low level of laccase production by natural microorganisms is the main bottleneck in its commercial application [15,16]. Therefore, much research is aimed at optimizing laccase expression in a simple heterologous expression system. In nature, white-rot fungi have been widely studied as one of the most effective producers of fungal laccase; So far, the heterologous expression of most laccases has been successful, for example, laccase genes have been cloned from *Fome lignosus* [17], *Pycnoporus cinnabarinus* [18], and various Trametes strains [19,20,21]. The difference between this study and the previous work mainly includes the following two points: 1) In previous studies, most recombinant laccases were used to degrade phenolic compounds, dye decolorization and degradation, etc., such as the study of Zhao [5], Nagai [22], Baldrian [23], etc. While in this study, recombinant laccase was used to degrade lignin, and the its ability was determined from qualitative and quantitative methods, which were not available in previous studies; 2) in the previous research on the degradation of lignin, the most studied was woody lignin, such as the research of Wei [24], Liu [25], etc., however, in this study, herbal lignin was used as the research object, to explore the effect of recombinant laccase on the degradability of herbal lignin.

In the 1980s, the yeast species *Pichia pastoris* was the first to assume the identity of a host organism and express a foreign protein. Since then, the *Pichia pastoris* expression system, using methanol as the sole energy source, has been rapidly disseminated and applied [26]. Due to its advantages of high stability, high reproduction rate, and high expression rate, the *Pichia pastoris* expression system is widely used for recombinant expression of exogenous proteins. To date, hundreds of exogenous proteins have been successfully recombined and expressed in *Pichia pastoris*, and this number is still increasing every year [27]. *Pichia pastoris* expression vectors are divided into two types: inducible expression vectors such as pPIC9, pHIL-S1, pPICZαA, and pYAM75P and constitutive expression vectors such as pHIL-D2, pHWO10, pGAPZ, and pGAPZα The inducible-vector promoter will only initiate expression in the presence of methanol. In contrast, the constitutive-vector promoter does not require any induction and has strong operability, high stability, and high safety. Therefore, we used the constitutive vector for heterologous expression of the laccase gene in *Pichia pastoris*.

In this study, the key gene for lignin degradation, *lac-2*, was screened from the transcriptome sequencing data of *Pleurotus ostreatus*, and the recombinant expression vector pGAPZαA-Lac-2 was constructed. After linearization, the expression vector plasmid was transformed by electroporation into *Pichia pastoris* X33 competent cells, and positive transformants were screened on yeast peptone dextrose sorbitol (YPDS) medium containing antibiotic Zeocin. Then, we measured the enzymatic activity by shake-flask fermentation in YPDS liquid medium. The recombinant laccase was purified, and its enzymatic properties were determined. Finally, the ability of recombinant laccase to degrade lignin in corn stover was determined by the experiment of color changing circle and lignin degradation experiment.

## 2. Materials and Methods

### 2.1. Strain, Plasmids, and Reagents

The white-rot fungus strain *Pleurotus ostreatus* was provided by the microorganism of Heilongjiang Province and cultured on potato dextrose Agar (PDA) medium at 25 °C. Corn stover was supplied by surrounding agricultural areas. *Escherichia coli* (*E. coli*) DH5α was purchased from Shanghai Weidi Biological Co., Ltd. Trizol extraction reagent, antibiotic Zeocin, the *Pichia pastoris* constitutive expression vector pGAPZαA, and the *Pichia pastoris* X-33 strains were purchased from Invitrogen. The restriction enzymes EcoRI, XbaI, and XhoI were purchased from New England Biolabs Inc. T4 DNA Ligase, 2×Taq PCR Mix, and the pMD18-T simple vector were purchased from Dalian TaKaRa Company. DNA marker was purchased from Guangzhou Dongsheng Biotechnology Co., Ltd. and the protein marker was purchased from ThermoFisher Scientific. The gel recycling kit and plasmid extraction kit were purchased from China Tianhe Biochemical Technology (Beijing, China) Co., Ltd. All other reagents needed in the experiments were manufactured in-house at analytical-grade purity.

### 2.2. Bioinformatics Analyses

The *lac-2* gene (submitted to GenBank, accession number: MT313303) was screened out of the *Pleurotus ostreatus* transcriptome database. The phylogenetic tree was constructed by the NJ (neighbor-joining) method using MEGA5.22 software, and the repeated calculation value was set to 1000 times [28]. Analysis of protein physicochemical properties of *Lac-2* gene of *Pleurotus ostreatus* using the protparam program provided by expasy proteomics server (www.expasy.ch). Predictive analysis of signal peptide cleavage sites using online software Signalp 3.0 (http://www.cbs.dtu.dk/services/signalp/) [29].

### 2.3. Amplification of lac-2 Gene and Connection with Cloning Vector

Total RNA was extracted from cultures of the *Pleurotus ostreatus* strain using TRIZOL reagent, and RNA concentration was measured by ultraviolet spectrophotometer, single-stranded complementary DNA (cDNA) was synthesized by reverse transcription, the method of reverse transcription test refered to the Guide to Molecular Cloning [30]. Specific primers were designed according to the nucleotide sequence of *lac-2* (generated by transcriptome sequencing): the forward primer GGAATTCTTTCAAGACGCTC contained an EcoRI digestion site (underlined), and the reverse primer CTCTAGAGTCAGGTCAGTAAGAGCAGGGGGGGG contained an XbaI digestion site (underlined). The amplified product was detected by 0.8% agarose gel electrophoresis, and the purified product was recovered from the gel. The PCR-purified *lac-2* gene was ligated with the cloning vector pMD18-T simple for 30 min at 16 °C. Then, the ligated product was transferred into *E. coli* DH5α and plated on Luria-Bertani (LB) media containing the antibiotic ampicillin (100 μg/mL), the method for transformation and ligation of *E. coli* DH5α was referred to the Guide to Molecular Cloning [30]. After this antibiotic-resistance screening, positive clones were selected and sent to Shanghai Shenggong Bioengineering Co., Ltd. for sequencing. The plasmid with the correct sequencing result was named pMD18-T-Lac-2.

### 2.4. Construction of Recombinant Protein Vector pGAPZαA-Lac-2

The recombinant plasmid pMD18-T-Lac-2 and the constitutive expression vector pGAPZαA were digested with restriction endonucleases EcoRI and XbaI, simultaneously. The target gene fragment and expression vector were digested with the gel recycling kit, and T4 DNA Ligase was used to ligate the recombinant plasmids at 16 °C for 18 h. The ligation products (8 µL) were transferred into 92 µL *E. coli* competent DH5α, transformed, and plated on LB medium containing Zeocin (25 μg/mL) [31]. The plates were then incubated upside down (inverted culture) at 37 °C. After 16–20 h, the monoclonal clones were identified as positive transformants by PCR and double restriction-enzyme digestion, and sent to Shanghai Shenggong Bioengineering Co., Ltd. for sequencing. The plasmids with the same sequence as the lac-2 gene transcripts were named pGAPZαA-Lac-2.

### 2.5. Electrotransformation and Screening of Positive Transformants

The recombinant plasmid pGAPZαA-Lac-2 and the empty vector pGAPZαA plasmid were simultaneously linearized by the restriction endonuclease XhoI. Then, 10–20 µg linearized pGAPZαA-Lac-2, and the control vector pGAPZαA were electrotransformed into *Pichia pastoris* X33. The electrotransformation was conducted with two electric shocks at 1150 V and 25 µF for 4.8 ms. Immediately after the electric shock, pre-cooled 1 M/mol sorbitol was added, and the cell suspension was incubated for 30 min at 30 °C in the dark. Then, the cell suspension was mixed with 100 µg/mL Zeocin, and different volumes (50 µL, 100 µL, 150 µL, and 200 µL) were plated on YPDS medium. Following incubation at 30 °C in the dark for 2–5 days to screen for Zeocin resistance, we isolated a single colony, and its recombinant DNA was extracted using a yeast genomic-DNA extraction kit. The positive yeast transformant was confirmed by colony PCR and named X33-Lac-2.

### 2.6. Liquid Fermentation Culture and Purification of Recombinant Laccase

The positive transformant X33-Lac-2 was inoculated into 5 mL yeast peptone dextrose (YPD) fermentation medium at 28–30 °C and shaken at 220 r/min for 18 h. The 2.0% (volume fraction) culture was re-inoculated into 50 mL YPD fermentation medium, after shaking at 28–30 °C for 72 h, the supernatant of the fermentation broth was collected by centrifugation, which was the crude enzyme solution. In the process of liquid fermentation, 1 mL culture was taken at 12 h, 24 h, 36 h, 48 h, 60 h, and 72 h, respectively, and centrifuged at room temperature for 6 min to extract crude enzyme solution, and SDS-PAGE electrophoresis was used to detect the expression of the protein in each time period. The crude enzyme solution was centrifuged at 6000 × g for 15 min at 4 °C and then added to the Q-Sepharose XL column pre-equilibrated with 50 mM Tris HCl equilibration buffer (pH 7.5), and separated by anion exchange chromatography. The same buffer containing 300 mM NaCl was used for elution. The fractions were monitored by absorbance at 280 nm and by laccase activity. The active components were collected and concentrated by ultrafiltration. The buffer was 50 mm tris (pH 7.5). This solution was added to a gel filtration column Sephacryl S-200 HR, and filtered with the same buffer containing 0.1 M NaCl. The protein-containing solution was maintained at –80 °C. After purification, the purity of the recombinant laccase was purified by SDS-PAGE analysis, and the protein concentration was determined by the Bradford method.

### 2.7. Characterization and Kinetic Analysis of Recombinant Laccase

Using citrate-phosphate as buffer (pH2.0–8.0), 2 mM 2,2’-azinobis-3-ethylbenzothiazoline-6-sulfonic acid (ABTS), 10 mM 2,6-Dimethoxyphenol (DMP), 20 mM 2-Methoxyphenol (Guaiacol) as a substrate, the reaction volume was 1 mL, and the recombinant laccase enzyme activity was determined after 5 min of reaction at 25 °C. The oxidation of ABTS was monitored at 420 nm, DMP oxidation was monitored at 468 nm, and guaiacol oxidation was monitored at 470 nm, the enzyme activity was expressed as IU. One unit of laccase activity is the amount of enzyme that oxidized 1 umol of substrate per minute. The optimum temperature and pH were determined by performing enzymatic assays at different temperatures (10–90 °C) and pH levels (2.5–8.0). The diluted enzyme solution was added to the citrate-phosphate buffer solution with different pH, and the pH stability of different strains was determined after incubation at 25 °C for 24 h. The dilute enzyme solution was cultured at different temperatures for 30 min, 60 min, 90 min, 120 min, 150 min, and 180 min to study the thermal stability of different strains at different temperatures. Under the optimal pH of each substrate, the kinetic parameters of different strains on three substrates were studied. All kinetic studies were repeated three times, and the kinetic data were fitted to a hyperbola using the Michaelis–Menten equation. 

### 2.8. Degradation of Lignin by Recombinant Laccase X33-Lac-2

The determination of the lignin degradation ability of recombinant laccase X33-Lac-2 can be divided into qualitative determination and quantitative determination. The qualitative measurement was carried out using a color-changing circle experiment. Hyun-Chae Jung et al. [32] believe that microorganisms that can produce color-changing circles on a guaiacol plate medium have the ability to degrade lignin. The crude enzyme solution of 10–20 uL recombinant laccase X33-Lac-2 was inoculated on guaiacol medium (potato dextrose agar medium + 0.02% guaiacol) and cultured for 3–5 days at 25 °C, empty plasmid yeast was used as blank control. The experiment was repeated for three times. The lignin degradation experiment was used for quantitative determination. The native *Pleurotus ostreatus*, native *Pichia pastoris*, empty pGAPZαA plasmid yeast, and recombinant laccase X33-Lac-2 were inoculated into corn stover culture medium (potato dextrose medium + 1% Corn stover) according to the same dosage. After 120 h of incubation at 25 °C, 120 r/min, drying, and weighing were carried out at 105 °C. Corn stover culture medium without inoculation was used as blank control. The experiment was repeated three times. The method for determining the degradation of lignin was carried out according to ‘CB/T2677.8-1994 Determination method of acid-insoluble lignin content of papermaking raw materials’ [33]. 

## 3. Results

### 3.1. Bioinformatics Analysis of Laccase Lac-2 Gene

The *Lac-2* gene of *Pleurotus ostreatus* is 1599 bp in length and contains 533 amino acids. Analysis of the gene by the program SignalP 3.0 predicted a signal peptide with a probability of 1.000 and the highest probability for a cleavage site between amino acid position 23 and 24. The physical and chemical properties of laccase Lac-2 protein were predicted and analyzed by ProtParam online software. The molecular weight of 533 amino acids was 56,796.72 Da, and the theoretical isoelectric point was 4.68. The predicted result of Grand average of hydropathicity (GRAVY) was 0.060; a GRAVY value above zero indicates that the protein is hydrophobic [34]. The phylogenetic tree was constructed with MEGA5.22 software (Figure 1). The similarity between *Pleurotus ostreatus* Lac-2 (MT313303) and *Florida Pleurotus laccase* MK38 (SNU32358.1) was 99.44%, The similarity between *Pleurotus ostreatus* Lac-2 (MT313303) and *Pleurotus ostreatus* bilirubin oxidase (BAA85185.1) was 99.72%, indicating laccase Lac-2 is closely related to the evolution of these two enzymes.

### 3.2. Heterologous Expression of lac-2 Gene in PICHIA Pastoris

After double enzyme digestion verification, the electrophoresis revealed a gene band of 1599 bp and a vector band of 3147 bp, which were consistent with our expected band sizes (Figure 2a). The recombinant plasmid was successfully linked to the expression vector pGAPZαA, which we named pGAPZαA-Lac-2. The recombinant expression plasmid pGAPZαA-Lac-2 and empty vector pGAPZαA were completely linearized, transformed into *Pichia pastoris* X33 by electrotransformation, and positive colonies were screened by Zeocin resistance. The recombinant transformant genomic DNA was extracted using the yeast genome extraction kit, and the yeast transformant genomic DNA was detected by colony PCR using the *Lac-2* gene-specific primer. The detection result is shown in Figure 2b, and the detection of about 1599 bp showed that the recombinant plasmid pGAPZαA-Lac-2 had been successfully transformed into *Pichia pastoris* X33, and the yeast transformant X33-lac-2 had been successfully constructed. 

### 3.3. Purification of Recombinant Laccase

During liquid fermentation, 1 mL of culture was taken every 12 h to prepare a crude enzyme solution, and the protein expression was detected by SDS-PAGE. The method to purify recombinant laccase from culture medium is shown in Table 1, the total recovery of the enzyme was 20%, and the specific activity was 151.89 U/mg. After the crude enzyme solution was concentrated and purified, the purified protein was detected by SDS-PAGE. As shown in Figure 3, there were clear protein bands at about 57 kDa at all time points, consistent with the predicted protein level (theoretical value 56.8). The longer the culture was left shaking in growth media, the higher the protein expression (lane 3–7 in Figure 3), while the control group X33-pGAPZαA did not secrete corresponding bands (lane 2 in Figure 3). The purified protein showed a single band (lane 1 in Figure 3), suggesting that the *lac-2* gene was successfully expressed in *Pichia pastoris* X33.

### 3.4. Analysis of Enzyme Activity and Enzymatic Properties of X33-Lac-2

The optimum pH and temperature of X33-Lac-2 with non-phenols (ABTS) and phenols (DMP, guaiacol) as substrates are shown in Figure 4. The optimum pH values of X33-Lac-2 for ABTS, DMP, and guaiacol are 3.0, 3.0, and 3.5, respectively (Figure 4a). The difference between the optimum pH is typical for laccases, and different substrates have different oxidation mechanisms. The optimal temperature of recombinant laccase was investigated under the optimal pH conditions of the three substrates. It was found that the optimum temperature of X33-Lac-2 was 50 °C regardless of the substrate conditions (Figure 4b), indicating that the optimum temperature of X33-Lac-2 was independent of the matrix [35].

The pH stability of all strains is shown in Figure 5. Between pH 2.5 and 4.0 (Figure 5a), the enzyme had the best stability, and the relative enzyme activity was more than 80%, while between pH 4.5 and 5.5, the relative enzyme activity could also be more than 60%, but between pH 6 and 8, the enzyme stability was relatively poor, so it can be seen that this enzyme was relatively stable under acidic conditions, which is consistent with that of the native *Pleurotus ostreatus*. Between pH 2.5–6 (Figure 5b), the relative enzyme activity of the native *Pleurotus ostreatus* was more than 60%, and between pH 6.5–8, the relative enzyme activity decreased rapidly. The X-33 *Pichia pastoris* and empty pGAPZαA plasmid yeast were only between pH 4–5.5 (Figure 5c,d), relative enzyme activity was greater than 60%.

Under the condition of pH 3.5, the thermal stability of all strains was studied within 3 h (Figure 6). The relative enzyme activity was over 80% when kept at 30–70 °C for 30 min (Figure 6a). After 3 h, the relative enzyme activity was over 60% at 30 °C and 40 °C, indicating that the recombinant laccase X33-Lac-2 had a wide temperature range, and relatively stable under low-temperature conditions. After being placed for 3 h, the relative enzyme activity of the native *Pleurotus ostreatus* was above 50% at 30 °C (Figure 6b), and the stability of other temperature conditions was poor. The relative enzyme activity of X-33 *Pichia pastoris* and empty pGAPZαA plasmid yeast at 30–50 °C for 60 min was more than 50% (Figure 6c,d), indicating that the two strains had better stability at low temperature.

Table 2 lists the kinetic parameters (K_M_, k_cat_) of each strain when ABTS, DMP, and guaiacol were used as substrates. When ABTS was used as the substrate, the catalytic efficiency of all strains was the highest. The K_M_ of all strains was lower than that of the native *Pleurotus ostreatus*. Compared with the native *Pleurotus ostreatus*, when using ABTS and DMP as substrates, the K_M_ of recombinant laccase decreased by 0.66 times and 0.80 times, respectively, and the catalytic efficiency was increased by 2.56 times and 1.39 times. When guaiacol was used as the substrate, the K_M_, K_cat,_ and catalytic efficiency of the native *Pleurotus ostreatus* were higher than that of the recombinant laccase.

### 3.5. Degradation of Lignin by Recombinant Laccase X33-Lac-2

It can be seen from Figure 7a that the recombinant laccase X33-Lac-2 produces a color-changing circle on the guaiacol medium, while the empty plasmid yeast does not produce a color-changing circle (Figure 7b), indicating that the recombinant laccase X33-Lac-2 has the ability to degrade lignin. Comparing the results of liquid fermentation (Table 3), it was found that the lignin content of corn stover treated by recombinant laccase X33-Lac-2 was the lowest, followed by the native *Pleurotus ostreatus*, both of which were lower than *Pichia pastoris* X-33 and empty pGAPZαA plasmid yeast. Compared with the blank control, the lignin degradation rate of recombinant laccase X33-Lac-2 was 18.36%, which was significantly different from other treatment groups (*p* <0.05).

## 4. Discussion

Laccase is recognized as one of the three major enzymes that can degrade lignin. The biodegradation of lignin by laccase can not only alleviate some environmental consequences of the paper industry, but also reduce the environmental impact of the large-scale development and utilization of stover [36]. The rate of laccase production by fungi is low in their natural state and any industrial application of laccase would quickly result in the demand overwhelming the supply. Therefore, in this study, a constitutive expression vector pGAPZαA was used to construct a highly efficient recombinant expression vector and to efficiently express laccase in *Pichia pastoris*. The *Pichia pastoris* expression system can make up for the shortcomings of a prokaryotic expression system, and protein expression can be increased 10–1000 times compared to the normal expression level [37,38]. Compared with induced expression vector methanol, expression vector pGAPZαA has low toxicity and low risk, avoiding the harm caused by methanol in practical application [39,40]. PCR detection of recombinant yeast DNA and protein electrophoresis of X33-lac-2 showed that *lac-2* gene had been successfully expressed. The specific activity of the recombinant laccase X33-Lac-2 was 151.89 U/mg, which is much larger than the specific activity of the recombinant laccase in the study of Huang et al. [15], but smaller than the specific activity of the recombinant laccase in the study of Xu et al. [41], the specific activity indicated that the purification yield of X33-Lac-2 was not outstanding in this study. But the recovery rate of X33-Lac-2 was 20%, which was significantly higher than the recovery rate of recombinant laccase in the study of Kittl et al. [31], the recovery rate is also higher than the recovery rate of recombinant laccase in the study of Xu et al., which indicates that X33-Lac-2 has a superior recovery rate in this study. In the future, breakthrough fluorescence can be achieved in large-scale production, by changing fermentation conditions and high-density fermentation technology. 

The optimal pH of X33-Lac-2 for the three substrates ABTS, DMP, and Guaiacol was between 3.0–4.0, which was consistent with the optimal pH of most fungal laccases [13]. In this study, the relative enzyme activity of X33-Lac-2 was above 80% between pH 2.5 and 4.0, and the enzyme stability was poor between pH 6.0 and 8.0. This enzyme is relatively stable under acidic conditions, which is different from most laccases currently known, because most laccases currently known are more stable between pH 6.0–7.0, and the enzyme activity will be reduced under more acid environment [42]. After incubating at 30–70 °C for 30 min, the residual enzyme activity of X33-Lac-2 can reach above 80%, indicating that the enzyme has high thermal stability and has great industrial application potential. In the process of culture, recombinant laccase X33-lac-2 can only produce one kind of enzyme that can degrade lignin, namely laccase. However, *Pleurotus ostreatus* strains can produce enzymes related to lignin degradation, such as manganese peroxidase and lignin peroxidase, which can degrade lignin in cooperation with laccase [43]. However, in this study, Potato Dextrose Medium + 1% corn stover was also used as the treatment condition, the degradation rate of *Pleurotus ostreatus* to lignin was only 14.05%, while the degradation rate of recombinant laccase X33-lac-2 to lignin was as high as 18.36%, the difference was significant (*p* < 0.05), indicating that the reason for the low degradation ability of *Pleurotus ostreatus* to lignin was the low activity of laccase production, which may be due to the recombinant yeast is more suitable for cultivation in liquid environment than *Pleurotus ostreatus*, its enzyme production is relatively good.

## 5. Conclusions

A preliminary bioinformatics analysis of the *lac-2* gene of *Pleurotus ostreatus* revealed that it was highly conserved over evolutionary time and had almost the same biological function as the high-homologous gene, which was helpful in predicting the sequence of related, new proteins and establishing inter-species relationships. In this study, the *Lac-2* gene of *Pleurotus ostreatus* was transferred into *Pichia pastoris* by genetic engineering, and the enzymatic properties of recombinant laccase X33-Lac-2 were further understood in a wide range of pH and temperature. The heat and acid resistance of recombinant laccase X33-Lac-2 indicated that it had good industrial application prospects. Under the condition of liquid fermentation of stover, the degradation rate of corn stover lignin by recombinant laccase X33-Lac-2 was higher than that of the native *Pleurotus ostreatus*, which indicated that it was feasible to improve the lignin degradation ability of stover by using genetic engineering bacteria. This experiment provides a reference value for the practical application of laccase gene, and also lays a foundation for the biodegradation, processing and reuse of crop stover.

## Figures and Tables

**Figure 1 microorganisms-08-00601-f001:**
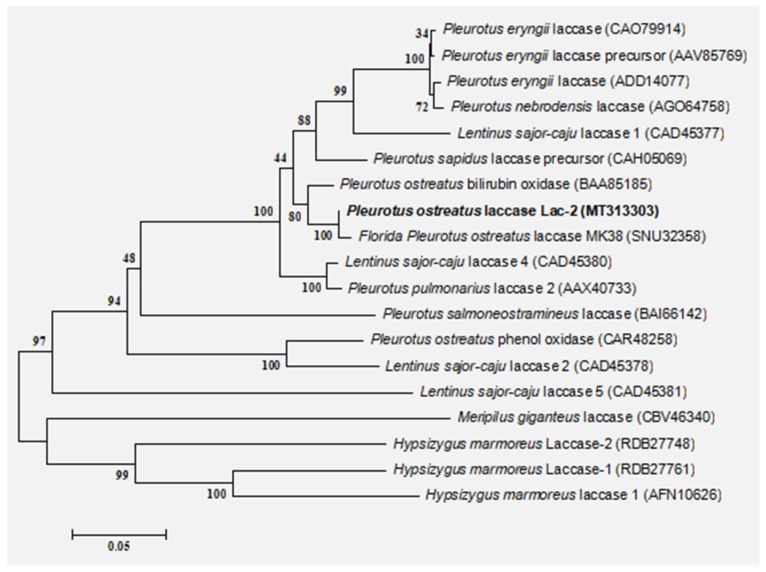
Phylogenetic tree of LAC-2 amino acid sequence. The neighbor-joining tree was constructed in Mega (Version 5.22) by using the Tamura–Nei substitution model and a thousand bootstrap replications.

**Figure 2 microorganisms-08-00601-f002:**
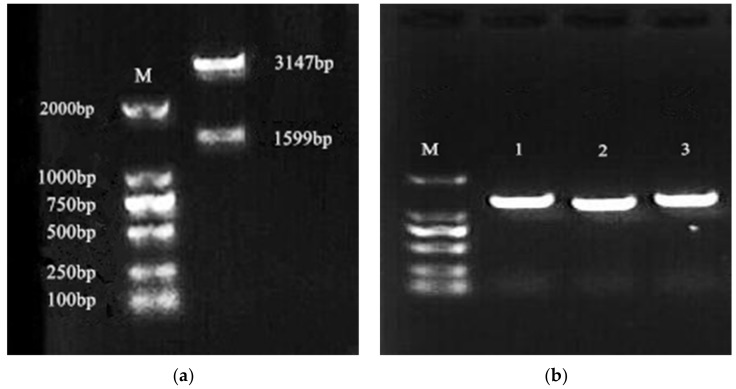
Double enzyme digestion identification and PCR verification of positive strains. (**a**) Double enzyme digestion identification results, lane ‘M’ is the DNA Marker; (**b**) detection results of PCR, lane ‘M’ is the DNA marker, lane 1–3 are the PCR product of X33-Lac-2.

**Figure 3 microorganisms-08-00601-f003:**
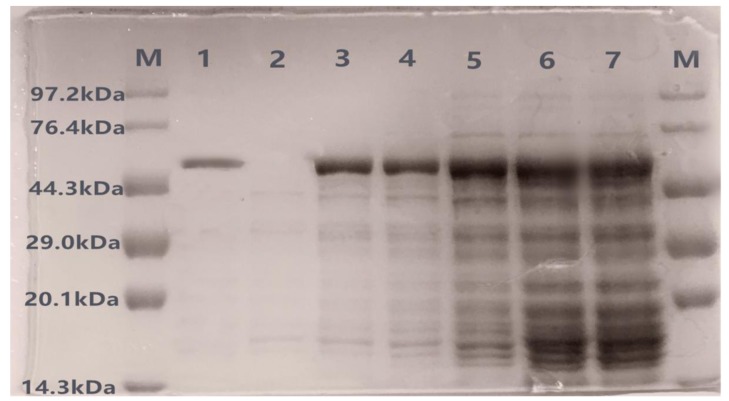
SDS-PAGE confirmation of recombinant protein X33-Lac-2. Lane ‘M’ is the protein maker. Lane 1 is purified recombinant laccase; lane 2 is empty vector ‘X33-pGAPZαA’; lane 3–7 are the protein expression band of 12 h, 24 h, 36 h, 48 h, and 60 h, respectively.

**Figure 4 microorganisms-08-00601-f004:**
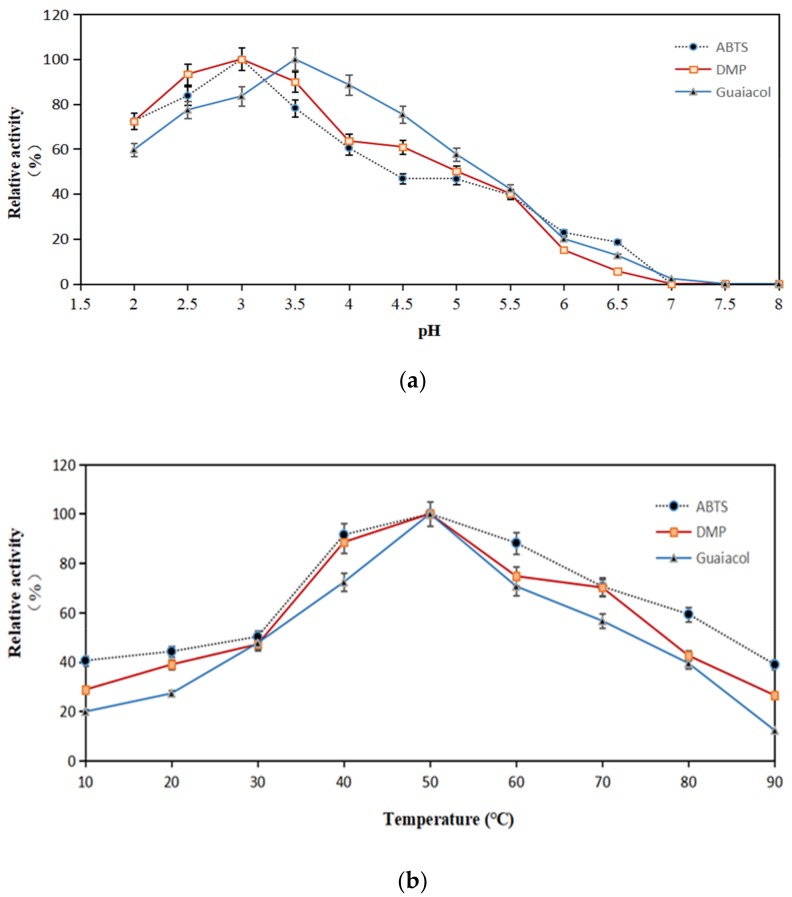
The optimum pH (**a**) and temperature (**b**) of X33-Lac-2. The effect of pH on the laccase activity was measured in citrate-phosphate buffer (pH 2.0–8.0) at 25 °C. The activity of 100% was that which was measured at the optimal pH. The effect of temperature on the laccase activity was measured at the optimal pH from 10 °C to 90 °C. The activity of 100% was that which was measured at the optimal temperature. Each point represents the mean ± SD (*n* = 3).

**Figure 5 microorganisms-08-00601-f005:**
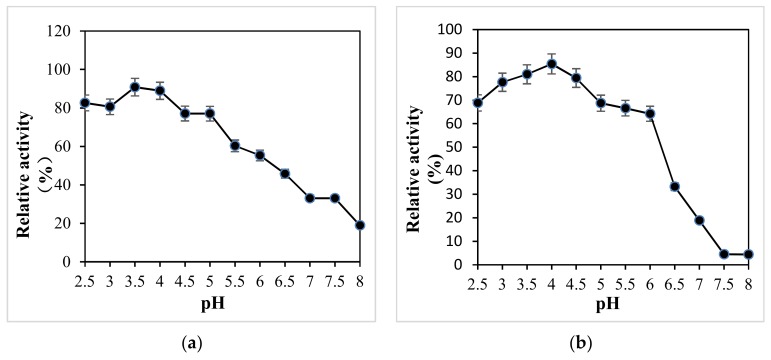
pH stability of all strains. (**a**) X33-Lac-2; (**b**) native *Pleurotus ostreatus*; (**c**) X-33 *Pichia pastoris*; (**d**) empty pGAPZαA plasmid yeast.

**Figure 6 microorganisms-08-00601-f006:**
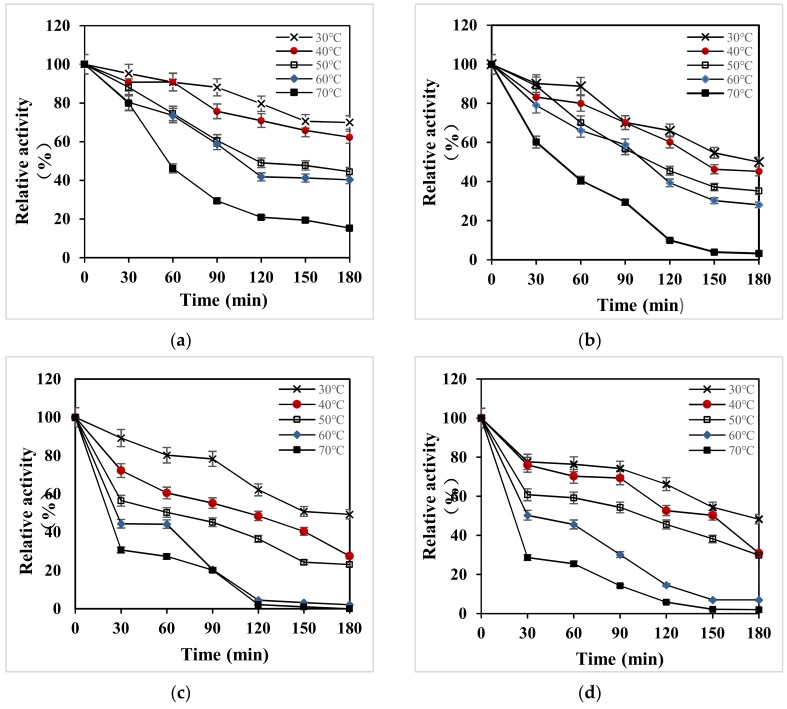
Temperature stability of all strains. (**a**) X33-Lac-2; (**b**) native *Pleurotus ostreatus*; (**c**) X-33 *Pichia pastoris*; (**d**) empty pGAPZαA plasmid yeast.

**Figure 7 microorganisms-08-00601-f007:**
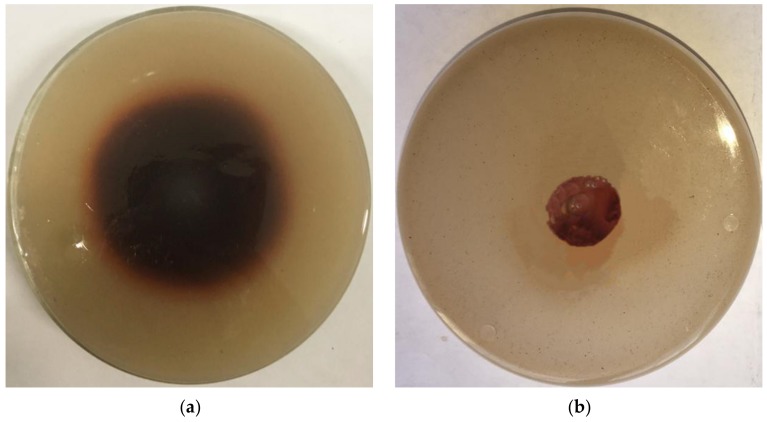
Color changing circle experiment. (**a**) Recombinant laccase X33-Lac-2; (**b**) Empty plasmid yeast.

**Table 1 microorganisms-08-00601-t001:** Purification of recombinant laccase.

Purification Step	Total Activity (U)	Total Protein (mg)	Specific Activity (U/mg)	Purification Factor	Yield (%)
Culture filtrate	5221	251.20	20.78	1.00	100
Ultrafiltrate	4563	135.23	33.74	1.62	87
Q-Sepharose XL	1770	28.47	62.17	2.99	34
Sephacryl S-200	1045	6.88	151.89	7.31	20

All recovery values are expressed in terms of activity units in the crude taken as 100%. The data obtained is the average of the three purification data.

**Table 2 microorganisms-08-00601-t002:** Kinetic parameters of each strain.

Strains	ABTS			2,6-DMP				Guaiacol	
	K_M_ (uM)	K_cat_ (S^−1^)	K_cat_/K_M_ (uM^−1^S^−1^)	K_M_ (uM)	K_cat_ (S^−1^)	K_cat_/K_M_ (uM^−1^S^−1^)	K_M_ (uM)	K_cat_ (S^−1^)	K_cat_/K (uM^−1^S^−1^)
*Pleurotus ostreatus*	20.6 ± 2.7	183.2 ± 22.2	8.89	99.0 ± 2.3	17.4 ± 3.1	0.18	452.0 ± 22.0	39.3 ± 2.7	0.086
X-33 *Pichia pastoris*	15.7 ± 3.9	123.0 ± 10.4	7.83	ND	ND	ND	395.7 ± 27.8	17.7 ± 4.4	0.045
Empty pGAPZαA plasmid yeast	14.5 ± 1.7	99.8 ± 10.1	6.88	ND	ND	ND	ND	ND	ND
Recombinant laccase X33-Lac-2	13.6 ± 2.9	310.8 ± 21.0	22.85	79.2 ± 18.3	19.8 ± 1.1	0.25	302.4 ± 26.0	15.9 ± 0.9	0.052

ND-Not Determined.

**Table 3 microorganisms-08-00601-t003:** The degradation rate of corn stover lignin by different strains.

Strains	Lignin Contents (%)	Lignin Degradation Rates (%)
*Pleurotus ostreatus*	11.93 ± 1.2524c	14.05 ± 1.3556b
X-33 *Pichia pastoris*	13.05 ± 1.3773b	11.71 ± 0.8534c
Empty pGAPZαA plasmid yeast	13.77 ± 0.9927b	9.99 ± 0.5434d
Recombinant laccase X33-Lac-2	9.13 ± 1.0222d	18.36 ± 1.0282a
Blank control	16.11 ± 2.0102a	ND

Different letters in each column represent significant differences (*p* <0.05). ND-Not Determined.

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
