# Peer review of "Expression of Pleurotus ostreatus Laccase Gene in Pichia pastoris and Its Degradation of Corn Stover Lignin"

_microorganisms, 2020, doi:10.3390/microorganisms8040601_

Round 1

Reviewer 1 Report

The manuscript is well written. However, before publication some improvements are necessary.

244 X33-Lac-2 was 50°C regardless of the substrate conditions (Figure 4b). indicating that

Both, pH stability and temperature stability shall be compared to other strains, as was done in Table 2.

Km for all three substrates and all strains shall be compared, as well.

Author Response

Response to Reviewer 1 Comments

The manuscript is well written. However, before publication some improvements are necessary. X33-Lac-2 was 50°C regardless of the substrate conditions (Figure 4b). indicating that Both, pH stability and temperature stability shall be compared to other strains, as was done in Table 2. Km for all three substrates and all strains shall be compared, as well.

Response :  First of all, thank you very much for your comments. I think your comments are very worthy of reference. So I added the corresponding experiment based on your opinion, and completed the experiment from April 9 to April 15, and added the corresponding results to the article, including:

1) Comparative test of pH stability and temperature stability of all strains involved in the experiment;

2) Comparative test of kinetic parameters of all strains involved in the experiment.

Thank you very much for your valuable suggestions.

Reviewer 2 Report

the article is a well-done job in the area of ligninolytic enzymes. however, this type of articles have been published before, please include in the introduction what thing makes this article novel or necessary. 

the major flaw of the article is the English, therefore, it needs to be improved before publication. the article has several misspellings, example of that is table 2 and the word "strans."

Author Response

Response to Reviewer 2 Comments

Point 1: The article is a well-done job in the area of ligninolytic enzymes. however, this type of articles have been published before, please include in the introduction what thing makes this article novel or necessary. 

Response 1: First of all, thank you very much for your comments. I think your comments are very worthy of reference. Based on your opinion, I added the innovation and necessity of this study in the preface of the article, and discussed the difference between this study and the published article.

Point 2: The major flaw of the article is the English, therefore, it needs to be improved before publication. the article has several misspellings, example of that is table 2 and the word "strans."

Response 2: Based on your opinion, I carefully checked the words of the article, and I have improved the English words in the article. Thank you very much for your valuable suggestions.

Round 2

Reviewer 2 Report

the article improvement is fair enough for publication